# Effects of Arbuscular Mycorrhizal Fungi on the Growth and Root Cell Ultrastructure of *Eucalyptus grandis* under Cadmium Stress

**DOI:** 10.3390/jof9020140

**Published:** 2023-01-19

**Authors:** Yuxuan Kuang, Xue Li, Zhihao Wang, Xinyang Wang, Hongjian Wei, Hui Chen, Wentao Hu, Ming Tang

**Affiliations:** State Key Laboratory of Conservation and Utilization of Subtropical Agro Bioresources, Guangdong Laboratory for Lingnan Modern Agriculture, College of Forestry and Landscape Architecture, South China Agricultural University, Guangzhou 510642, China

**Keywords:** cadmium stress, arbuscular mycorrhizal fungi, *Eucalyptus grandis*, cellular ultrastructure

## Abstract

*Eucalyptus grandis* (*E*. *grandis*) has been reported to form a symbiosis with arbuscular mycorrhizal fungi (AMF), which plays an important role in improving plant tolerance of heavy metal. However, the mechanism of how AMF intercept and transport cadmium (Cd) at the subcellular level in *E*. *grandis* still remains to be researched. In this study, a pot experiment was conducted to investigate the growth performance of *E*. *grandis* under Cd stress and Cd absorption resistance of AMF and explored the Cd localization in the root by using transmission electron microscopy and energy dispersive X-ray spectroscopy. The results showed that AMF colonization could enhance plant growth and photosynthetic efficiency of *E*. *grandis* and reduce the translocation factor of Cd under Cd stress. After being treated with 50, 150, 300, and 500 μM Cd, the translocation factor of Cd in *E*. *grandis* with AMF colonization decreased by 56.41%, 62.89%, 66.67%, and 42.79%, respectively. However, the mycorrhizal efficiency was significant only at low Cd concentrations (50, 150, and 300 μM). Under 500 μM Cd concentration condition, the colonization of AMF in roots decreased, and the alleviating effect of AMF was not significant. Ultrastructural observations showed that Cd is abundant in regular lumps and strips in the cross-section of *E*. *grandis* root cell. AMF protected plant cells by retaining Cd in the fungal structure. Our results suggested that AMF alleviated Cd toxicity by regulating plant physiology and altering the distribution of Cd in different cell sites.

## 1. Introduction

Cadmium (Cd) accumulation in soil has become a serious global concern with pronounced harmful effects on water, agricultural productivity, and human health [1]. In dealing with Cd pollution, people developed phytoremediation technology, which uses metal-accumulating plants to remove toxic metals from contaminated soil [2,3]. However, hyperaccumulator plants are mostly herbaceous [4]. Herbaceous’ vulnerability to soil moisture, salinity, and pH greatly limited its practical application [5]. It is crucial to find a woody plant with high biomass, rapid growth, and good economic returns that can be used for the phytoremediation of heavy-metal-contaminated soil [6]. *Eucalyptus grandis* (*E. grandis*) is one of the most widely distributed fast-growing timber forests with a long taproot, a strongly developed root system, and large biomass [7]. Some researchers reported that arbuscular mycorrhizal fungi (AMF) symbiosis with *E. grandis* can enhance the heavy metal resistance of plants [8,9]. Therefore, *E. grandis* contains great potential and ecological value as a bio-remediation tree for heavy-metal-contaminated soil [10]. However, the extent of Cd tolerance in *E. grandis* and how Cd is transported at the subcellular level remain to be investigated.

AMF can form a symbiotic relationship with 85% of plant roots, improves the resistance of host plants towards heavy metal toxicity, and enhances plant stability under stress [11,12]. AMF assisting plant remediation is an ecological, economic, and stable green remediation technique that shows great potential in the ecological restoration of heavy-metal-contaminated soil [13]. Nafady and Elgharably [14] confirmed that after inoculation with mycorrhizal fungi, especially with *Rhizophagus aggregatus*, Cd, Pb, Zn, and Cu concentrations in branches are significantly reduced through the release of glomalin. Babadi et al. [15] found that AMF acts as a rhizosphere protection mechanism to increase the concentration of glomalin, reduce Cd transport in sorghum roots, and effectively limit Cd content in plant tissues. Our previous study indicated that AMF increases the resistance of *E. grandis* to high-Zn stress by improving nutrient uptake and regulating Zn uptake at the gene transcription level [16]. Moreover, numerous studies showed that AMF inoculation can significantly alleviate the ultrastructure damage of host plant cells caused by heavy metals. Turnau et al. [17] used electron energy loss spectroscopy to study the distribution of heavy metal elements in the roots of the *Pteridium aquilinum* (L.) Kuhn found that the cytoplasm of the mycorrhizal fungus contains more Cd, Ti, and Ba than that of the host cells of the fern and named this combination the “filtering” mechanism. Jerusa et al. [18] found that inoculating *Rhizophagus clarus* and mixed AMF (*Acaulospora morrowiae*, *Gigaspora albida,* and *Rhizophagus clarus*) significantly alleviated As stress to the ultrastructure of leaves and root cells of *Leucaena leucocephala* (Lam.). Wu et al. [19] provided evidence for Cr immobilization on the fungal surface and in fungal structures in mycorrhizal roots at a cellular level and thus unraveled the underlying mechanisms by which AM symbiosis immobilizes. Xu et al. [20] demonstrated that *Funneliformis mosseae* could keep funneling Pb ions inside their intraradical mycelia to alleviate the Pb toxicity to the root cells of *Sophora viciifolia*. Huang et al. [21] found that AMF colonization enhanced Cd tolerance of *Phragmites australis* (Cav.) Trin. ex Steud. via Cd uptake and distribution at subcellular levels, and AMF plays a major role in the enhancement of Cd tolerance. Chen et al. [22] clarified the process of direct absorption and transport of Cd by AMF and revealed in detail the spatial distribution of Cd in mycorrhiza after Cd absorption by fungi, providing direct evidence for the intraradical immobilization of Cd absorbed by AMF. However, the concentration sensitivity of mycorrhizal eucalyptus to Cd transport processes at the cellular level and whether AMF can alleviate ultrastructural changes of woody plant root by Cd toxicity are unclear.

In the present study, we focused on the growth characteristics of *E. grandis* symbiosis with *Rhizophagus irregularis* (*R. irregularis*) under different Cd concentration conditions and the changes in root cell ultrastructure of *R. irregularis* -*E. grandis* symbionts. The aims of our investigation include the following: (1) to elucidate how AMF inoculation affected the growth parameters, photosynthesis parameters, Cd absorption, and root cell ultrastructure of *E. grandis* under different Cd concentrations; (2) to explore the distribution sites and characteristics of Cd in non-mycorrhizal/mycorrhizal *E. grandis* roots by TEM-EDS. The findings from our study may offer a theoretical foundation for the mechanism of Cd toxicity alleviation by AMF and provides support for the application of *E. grandis* in heavy metal soil remediation.

## 2. Materials and Methods

### 2.1. Plant Materials and AMF Colonization

*E. grandis* seeds were surface-disinfected with 3% sodium hypochlorite for 20 min and rinsed with sterile distilled water. The seeds were germinated on LA medium at 25 °C in the dark until the hypocotyls elongated about 1 cm and then transferred to the growth chamber controlled at 26/22 °C; 60% relative humidity; a 16 h/8 h light/dark cycle; 650 mmol m^–2^ s^–1^ photosynthetic active radiation (PAR). After two weeks of cultivation, selected seedings with the same growth and transplanted into small plastic pots (diameter 10 cm and height 9 cm). *R. irregularis* DAOM 197198 was selected as the experimental AMF and was obtained from the forestry and landscape architecture College of South China Agricultural University (Guangzhou, China). *Zea mays* L. was used as a host plant for propagation. After propagation, the spore agent obtained by sucrose gradient centrifugation was used as inoculum, and about 400 spores were inoculated in each pot of seedlings.

### 2.2. Experiment Design

Pot experiments were undertaken at the State Key Laboratory of Conservation and Utilization of Subtropical Agro-Bioresources (Guangzhou, China). A mixture of 0.6 mm sand and vermiculite was sterilized at 121 °C for 2 h to serve as the culture medium. The range of tolerance of *E. grandis* was obtained through preliminary experiments and previous data from our research group. Therefore, our experiment included five levels of Cd concentration: 0, 50, 150, 300, and 500 μM; two inoculation treatments: non-mycorrhizal treatment (NM) and AMF inoculation (AM). Each treatment was replicated six times, resulting in a total of 60 rectangular pots. The inoculation containing 400 AMF spores was added to the substrate of the AM group, and the same amount of spore-free filtrate was applied to the control group. After AMF inoculation, all seedlings were normally watered 30 mL each day, and LA nutrient solution with low phosphorus content (contains one-tenth of phosphorus) was poured once each 3 days, 50 mL per pot. After 45 days, confirmed the colonization, then started heavy metal treatment. Different concentration of cadmium chloride (CdCl_2_) was applied once every 5 days, 50 mL each time, for a total of 8 times. After 40 days of heavy metal stress, the samples were collected for the determination of relevant indicators.

### 2.3. Growth Indicators Determination

#### 2.3.1. Biomass

Five seedlings were randomly selected for each treatment, and the surfaces of roots, stems, and leaves were gently washed with tap water to drain the surface moisture. The fresh weight of the root and shoot of *E. grandis* was measured by electronic balance (accurate to 0.001 g). Part of the weighed plant samples was placed at 105 °C for 10 min, then continuously placed at 70 °C for 72 h until constant weight and measured the dry weight of the shoot and root parts of the plant.

#### 2.3.2. Mycorrhizal Colonization

The improved trypan blue staining method was used for staining using the method [23]. The grid-line crossing method [24] was used to measure the colonization of AMF in roots, and related indexes such as total colonization rate of roots, arbuscular colonization rate, and vesicle colonization rate were recorded.

#### 2.3.3. Chlorophyll Fluorescence Parameters

In total, 5 plants were randomly selected from each treatment, and dark adaptation was conducted for 30 min. The 5th–6th leaves from top to bottom were selected and determined by a dual-channel chlorophyll fluorescence analyzer (Dual-PAM-100, WALZ, Bavaria, Germany). These chlorophyll fluorescence parameters were adopted from the following equations defined by Rascher et al. [25]. ΦPSII = (F_m_′ − F′)/F_m_′; F_v_′/F_m_′ = (F_m_′ − F_0_′)/Fm′, where F_m_ and F_0_ are the maximal and minimal fluorescence values of dark-adapted leaves after 30 min, respectively, F_v_ is the variable fluorescence, Fm′ is the maximal fluorescence of the light-adapted leaves. ETR is the electron transport rate, and qP is the photochemical quenching coefficient; these two indicators are directly measured by the instrument.

#### 2.3.4. Cd Concentration Analysis

Parts of the roots and leaves were washed with 10 mmol L^−1^ EDTA (pH 8.0) and rinsed with ddH_2_O_2_ to remove the metal ions from the root surface [26], oven-dried at 105 °C for 30 min, followed by 70 °C until constant masses, weighed and ground with a stainless-steel mill, then passed through a 0.1 mm nylon sieve. About 0.1 g of the plant sample was digested using the HNO_3_–HClO_4_ (3:1) method [27]. The digested solutions were washed with ddH_2_O_2_ in a 15 mL centrifuge tube. Cd quantification was performed with an Inductively Coupled Plasma Emission Spectrometer (prodigy 7, Leeman, Hudson, NH, USA). The following formulae were used to calculate the key parameter that was analyzed [28]. Translocation factor (TF) = Cd _shoots_/Cd _roots_. Cd _shoots_ denotes the Cd concentration in shoots, while Cd _roots_ denotes the Cd concentration in roots.

#### 2.3.5. Cd Localization in Roots of *E. grandis*

Ultrathin section from the NM and AM-treated *E. grandis* root cell was negatively stained with 2% uranyl acetate and 1% lead citrate. The cross-section of AMF-*E*. *grandis* symbiont cell and the distribution of Cd in *E*. *grandis* roots were examined under a Field Emission Transmission Electron Microscope (Talos F200S, FEI company, Hillsboro, OR, USA).

### 2.4. Statistical Analysis

SPSS 22.0 statistical program (SPSS Inc., Chicago, IL, USA) was used to statistically analyze the growth parameters, chlorophyll fluorescence parameters, AMF colonization, and Cd concentration in NM/AM. All data met the normality and homogeneity of variance assumptions and were analyzed by ANOVA with means comparisons according to Duncan’s multiple range test (*p* < 0.05). For the biomass, chlorophyll fluorescence parameters, and Cd concentration of plants, two-way ANOVAs were used to assess the effects of “AMF” and “Cd treatment”. Graphs were prepared using SigmaPlot version 12.3 (Systat Software Inc., San Jose, CA, USA).

## 3. Results

### 3.1. Effects of Different Cd Concentrations on AMF Colonization

Different Cd concentrations affected the AMF colonization on *E*. *grandis* roots. The proportion of arbuscule was relatively high under non-stress conditions. The proportion of vesicles and hyphae increased with the increase in Cd concentration. Under 500 μM Cd concentration condition, a large amount of degraded arbuscule was observed (Appendix A). Compared with the non-stress group, the total colonization rate increased significantly when Cd concentration was 50, 150, and 300 μM, which reached 67.40%, 68.67%, and 66.66%, respectively. However, when the Cd concentration reached 500 μM, the total colonization rate decreased to 63.80% (Figure 1A). The colonization rate of arbuscule decreased with the increase in Cd concentration. Compared with the non-stress group, the colonization rate of arbuscule decreased by 14.40%, 39.70%, 53.00%, and 52.68% under 50, 150, 300, and 500 μM Cd concentration conditions (Figure 1B). The colonization rate of vesicles increased significantly with the increase in Cd concentration and reached the maximum value at 500 μM, which was 156.28% higher than that under non-stress conditions (Figure 1C).

### 3.2. Effects of AMF on E. grandis Growth Parameters under Different Cd Concentrations

We found that under Cd stress, *E*. *grandis* could grow under all concentration conditions without AMF colonization, but plants showed stunting, such as wilting and red leaves. In the high Cd concentration treatment (500 μM) group, the leaves of the plant even showed a lot of litter (Appendix A). The AM group exposed to Cd stress also showed Cd toxicity, such as decreased height and red leaves, but the toxicity degree was significantly less than the NM group (Appendix A).

After 40 days of Cd treatment, the dry weight of both shoots and root of *E*. *grandis* decreased significantly with the increase in Cd concentration, and AMF could significantly alleviate the reduction of biomass. The mycorrhizal efficiency was most significant under 300 μM Cd concentration condition. The dry weight of shoots and roots in the NM group decreased by 47.84% and 49.59% compared with the non-stress group. However, the dry weight of shoots in AM group only reduced by 34.97% and 36.51% compared with the non-stress condition (Figure 2). When the concentration of Cd increased to 500 μM, there was no significant difference in dry weight between the NM and AM groups, and the mycorrhizal efficiency of AMF was insignificant. The effects of AMF, Cd reached significant levels (*p* < 0.05) on all tested biomass of root and shoot, but their interactions showed no significant effect on those parameters.

### 3.3. Effects of AMF on E. grandis Photosynthesis Parameters under Different Cd Concentrations

Under Cd stress, AMF could significantly increase ΦPSII, Fv′/Fm′, ETR, and qP (Figure 3). AMF inoculation significantly increased the PSⅡ under non-stress and low Cd stress (50 and 150 μM); compared with the NM group, AM group increased by 43.33%, 50.69%, and 65.09%, respectively (Figure 3A). Fv′/Fm′ decreased with the increase in Cd concentration; it decreased by 26.88% and 31.36% Cd concentration of 300 μM and 500 μM compared with that under non-stress condition, and AMF inoculation significantly increased Fv′/Fm′ by 37.01% and 43.71% (Figure 3B). Meanwhile, AMF inoculation significantly increased the ETR under non-stress and low Cd stress (50, 150 μM), compared with the NM group, AM group increased by 43.19%, 52.07%, and 64.67%, respectively (Figure 3C). The qP was significantly increased by AMF inoculation under non-stress and low concentration Cd stress (50 and 150 μM); AMF inoculation significantly increased the qP, compared with the NM group, AM group increased by 39.49%, 23.95%, and 55.58% (Figure 3D). The effects of AMF and Cd reached significant levels (*p* < 0.05) on all tested chlorophyll fluorescence parameters, but their interactions showed no significant effect on those parameters.

### 3.4. Effects of AMF on the Cd Absorption of E. grandis

There were differences in total Cd concentration in *E*. *grandis* and Cd concentration in shoot and root under different Cd concentration conditions (Figure 4). The Cd concentration in the plantlet increased significantly with the increase in Cd concentration, and the effect of Cd reached significant levels (*p* < 0.05) on tested total Cd concentration in *E*. *grandis*, but the effect of AMF their interactions showed no significant effect on those parameters (Figure 4A). However, AMF inoculation significantly increased Cd concentration in the root while significantly reducing Cd concentration in shoot; the effects of AMF and Cd reached significant levels (*p* < 0.05) on tested Cd concentration in both shoot and root (Figure 4B,C). Under 300 μM Cd concentration condition, the effect of AMF inoculation was the most significant, and the concentration of Cd in the shoot was reduced by 64.08% (Figure 4B). AMF reduced the translocation factor of Cd. After being treated with 50, 150, 300, and 500 μM, the translocation factor in AM group decreased by 56.41%, 62.89%, 66.67%, and 42.79%, respectively (Figure 4D).

### 3.5. Effects of AMF and Cd Stress on E. grandis Root Cell Ultrastructure

We found that under 500 μM Cd concentration condition, the morphology of NM root cortex cells changed, most of the cells showed irregular strip shape, some cells burst (c in Figure 5) or even collapsed (b in Figure 5), and a large number of gray black and dark black cells (a in Figure 5) appeared compared with the non-stress group, and that the dark black lipids were concentrated in the lateral cells of the stele and the lateral cells of the endodermis of the *E. grandis* roots. We suspected this deposition site was Cd. Compared with the NM group, it can be seen that gray black cells in AM group decreased obviously, and we found the arbuscular structure of AMF entered the cells of the *E*. *grandis* root system and were able to occupy the whole cell (d in Figure 5), which was relatively intact compared to other cells that are deformed, ruptured or even collapsed. Hypha of AMF could also be observed under field emission transmission electron microscopy.

At the subcellular level in *E*. *grandis* root without AMF colonization, the cell structure was obviously incomplete, cell membrane breakage resulting in cell wall collapsed and tissue breakage (Figure 5C). Cd toxicity has a great effect on NM cell structure. We found AMF inoculated in *E*. *grandis* root, and the symbiont cell was complete and regular compared with Cd-poisoned cells; AMF could effectively alleviate cell deformation and collapse caused by Cd toxicity (Figure 5D). In the local figure of fungal structures in *E*. *grandis* roots, it can be seen that 5D was arbuscular cell, and 5E and 5H were hypha structures. Hexagonal crystals with regular shape were deposited around the arbuscular cells (g in Figure 5), which we suspected to be complex precipitation; the sites indicated by arrows h and j were suspected of Cd deposition in hypha.

We performed energy spectrum analysis on the parts of NM *E*. *grandis* cells, which were suspected to be Cd deposition (Figure 6A) and collapsed cells (Figure 6E). The results are shown in Figure 6D,H; the occurrence of a Cd peak indicates that there are indeed a large number of Cd deposits in these two sites, and the distribution range of Cd is shown in Figure 6C,G. We can clearly see that Cd is uniformly concentrated in the gray and black parts and the collapsed cell wall under TEM-EDS; the results showed that the broken and collapsed cells in NM roots were indeed caused by the toxic effect of Cd, and we can preliminarily speculate that the deposition position of Cd in the root cells without AMF inoculation is in the dark black cells in the section. However, the EDS result of AM root showed that Cd was abundant in vesicle cells and existed as regular strips (Figure 6K); the degree of cell collapse was obviously less than the NM group (Figure 6I). With AMF colonization, Cd-poisoned cells maintained relatively intact cell morphology and structure. These results suggest that AMF could help *E*. *grandis* root cells retain Cd, thus reducing the toxicity of Cd to *E*. *grandis* cells.

In addition, we found the distribution and form of Cd in cross sections of *E*. *grandis* root by analyzing the areas of Cd enrichment sites in cells treated with Cd at 500 μM under field emission transmission electron microscopy. The bright patches are the locations of Cd in the energy spectrum. The results of the EDS analysis showed that peaks of C, O, Os, Cu, and Cd appeared (Figure 7C,F). The samples’ biological nature resulted in the appearance of C and O peaks in the figure. The carrier network of transmission electron microscope uses the Cu network, so the Cu peak appears. Because the sample was stained with osmium, the Os peak appeared. The appearance of Cd peaks proves that these bright patches are indeed made of Cd, and Figure 7B,E show the form of Cd; Cd are abundant in regular lumps and strips.

## 4. Discussion

### 4.1. AMF Improved the Growth and Photosynthetic Efficiency of E. grandis under Cd Stress

In the present study, we found that the biomass of *E*. *grandis* decreased with the increase in Cd concentration, Cd toxicity inhibited plant growth, and AMF colonization strongly enhanced the biomass of *E*. *grandis* under Cd stress except for 500 μM. AMF are widely believed to support plant establishment in soils contaminated with heavy metals because AM symbiosis with plants promotes the growth of the host [29,30]. In this study, *R. irregularis* inoculation enhanced plant development under Cd stress, which was manifested by higher shoot and root biomass.

Chlorophyll fluorescence parameters are important indicators to show plants’ responses and adaptations toward heavy stress [31]. Our results indicated that *E*. *grandis* showed significantly lower F_v_′/F_m_′ and qP under Cd stress, which infers the negative impact of Cd on photosynthesis. Moreover, the leaves of *E*. *grandis* inoculated with *R. irregularis* maintained higher ΦPSII, ETR, and qP values under Cd stress. A previous study showed that the loss of photosynthesis performance ΦPSII was caused by the decrease in F_v_′/F_m_′ and qP, which indicates a reduced consumption of NADPH and ATP in the Calvin cycle due to reduced demand of sugars in other plant parts caused by Cd or inhibition of enzymes of the Calvin cycle [11]. Reduction in the quantum yield of photosystem II and electron transport rate and the increase in the non-photochemical quenching and steady-state chlorophyll fluorescence showed that the capture of light energy by plants was missed out in the form of the non-fluorescence quenching [9]. In this research, F_v_′/F_m_′, ΦPSII, ETR, and qP in *E*. *grandis* increased after *R. irregularis* inoculation compared with non-mycorrhizal *E*. *grandis* under cd stress, suggesting that AMF colonization alleviated the cd stress in *E*. *grandis* while significantly increasing photosynthetic protection.

High concentrations of metals can inhibit AMF spore germination and extra-root mycelial growth, thus affecting the colonization of AMF in plant roots [32]. However, some researchers have reported different results. For example, the colonization rates of *Rhizophagus intraradices* to wild tobacco were increased from 14 to 81% when soil Zn levels were increased from 0 to 250 mg kg^−1^ [33]. Due to the capacity of the AMF isolate to inhabit contaminated soils as well as mycorrhizal dependency, the mycorrhizal colonization observed can be partly explained [21]. AMF has a certain tolerance to heavy metal stress, and a high concentration of Cd stress has a certain impact on the growth and development of AMF. Thus, high concentrations of heavy metals decrease the colonization of AMF in plant roots, but such findings may differ due to a wide variety of reasons, such as the compatibility of AMF species and plants, element concentration of metal and oxidation state, and environmental conditions [34,35]. Our results indicated that Cd treatment (50, 150, 300 μM) significantly increased AMF colonization compared with non-Cd treatment. The reason for this may be the vesicles and hyphae of AMF are used for chelating heavy metals after the addition of Cd. While under a Cd concentration of 500 μM, high concentrations of Cd limited the growth of AMF, root colonization decreased, and AMF had no significant effect on growth parameters and photosynthetic efficiency.

### 4.2. AMF Mobilizes the Distribution and Transportation of Cd in E. grandis

The mycorrhizal efficiency resulted in a biomass allocation mechanism with a preference for the root system, ensuring tolerance, enrichment, and immobilization of the root system to toxic trace element contamination [36,37]. AMF was demonstrated to affect heavy metals bioavailability in the host plants and promote HMs retention in the roots, improving shoot biomass by restricting translocation to the aerial parts [21,38]. We observed that although there was no significant difference in the total Cd content between mycorrhizal and non- mycorrhizal plants, the Cd concentration in the roots of mycorrhizal plants was significantly higher than that in the shoots, indicating that AMF colonization significantly inhibited the migration of Cd from the roots to the shoots, thus limiting the accumulation of Cd in the photosynthetic tissues and effectively alleviating the toxic effect of Cd on the shoots.

On the other hand, AMF colonization has different effects on the distribution and transportation of heavy metals in plants [39,40]. Qiao et al. [41] showed that after being inoculated with *Funneliformis mosseae, Medicago sativa* L. had higher Co and Pb concentrations, and AMF inoculation promoted HMs translation to shoot. Citterio et al. [42] reported the influence of *Funneliformis mosseae* inoculation on the Cd, Ni, and Cr uptake of *Cannabis sativa* L. under heavy metal and found that the concentration of heavy metal was not affected by inoculating *F. mosseae*. Cui et al. [43] found *R. irregularis* effectively impacted the partitioning of Mg, Cu, and Zn accumulation and partition, including increased Mg and decreased Cu and Zn relative concentrations in shoots of soybean. Hence, the effects of AMF inoculation on the distribution and transportation of heavy metal are associated with AMF and plant species [20]. In our study, *R. irregularis* played an important role in regulating the distribution and transport of Cd in *E*. *grandis*, protecting the photosynthetic system by accumulating more Cd in the root and reducing Cd transport to the shoot. Therefore, *E*. *grandis* symbiosis with *R. irregularis* is a feasible strategy for the remediation of Cd-contaminated soils.

### 4.3. Damage of Cell Ultrastructure in E. grandis Root by Cd and Positive Effect of AMF Colonization

Plants’ detoxification to heavy metal by fixing heavy metal particles in cell structures (vacuoles, cell walls), which are not involved in important physiological and biochemical processes [44]. Here, we investigated the altered ultrastructure of *E*. *grandis* cells and hypothesized that AMF jointly alleviate Cd toxicity by altering the distribution of Cd in different cell sites. We found that Cd are abundant in regular lumps and strips in the cross-section of *E*. *grandis* root cell; Cd was present in some intact vacuoles and in the walls of some collapsed cells. This result is similar to the findings of Kiran et al. [45], who found that Cd in the root system of *Brassica rapa subsp.* chinensis L. mostly exists in the vacuole or attached to the cell wall and exhibited exudates of Cd-dense material. Cd was observed in the roots of mycorrhizal plants in the form of aggregated regular strips and blocks attached to the cell walls or in vacuoles, which reduced the toxicity of Cd to the plant cells. Meanwhile, we found Cd possibly induced alterations in the ultrastructure of endodermal and stele cells. In the entire cortex of *E. grandis*, especially in cells located closer to the stele, there appeared a large number of cells containing dark black material (stained with uranium dioxy acetate and lead citrate), and these cells showed deformation or even rupture, and they showed electron-dense cytoplasm, which was preliminarily speculated as the deposition location of Cd, and we speculated that the black deposits were lipids and that the dark black lipids were concentrated in the lateral cells of the stele and the lateral cells of the endodermis of the *E. grandis* roots. These results agree with the study of Katarzyna Głowacka et al. [46]; they noticed in almost all tissues of the root was an increased number of oil bodies, while the oil bodies were mainly observed in the cytosol of the vascular parenchyma cells, the cytoplasm of other stele cells, and in the cortical cells treated with Cd. It should be noted that high concentrations of Cd in the root epidermis have been reported in previous studies [33,47]. A large amount of Cd might accumulate between the epidermis and outermost layer of the cortex, and a higher concentration of Cd within the stele implies that a considerable fraction of the Cd absorbed by the root has entered the vascular bundle in the root [45]. In addition, our observations were similar to Katarzyna Głowacka et al. [46], the roots exposed to Cd, a strong H_2_O_2_–DAB dependent reaction was also seen in the entire cortex, especially in cells located closer to the stele. Thus, Cd ions tend to accumulate in the cells located closer to the stele in the cross-section of the *E*. *grandis* root cell and exist in regular lumps and strip cells, which we suspected to be lipid-containing structures. The lightening of the color of this part of the cells under TEM after inoculation with AMF also indicates that AMF symbiosis reduced Cd transport through the mid-column to the above-ground part.

A previous study reported that AMF improves heavy metal tolerance in plants due to its ability to sequester heavy metal ions in their intraradical mycelia [20]. It has also been shown that this toxic metal was mainly retained in the fungal structure, particularly in the arbuscules, and did not appear to be delivered to plant cells [22]. Our results indicated that cells containing lipids material significantly decreased in the root inoculated with *R. irregularis*, and we found that the deposits of Cd were mainly concentrated around the arbuscular structure, in the form of a regular lens, or in the hypha structure. Thus, at the subcellular level, *E*. *grandis* symbiosis with *R. irregularis* could protect the root cell structure of the plant by concentrating Cd around the arbuscular structure or in the hypha structure and effectively maintaining the integrity of the nearby cell morphology. This may be an effective strategy for *R. irregularis* to enhance the heavy metals tolerance of *E*. *grandis.*

## 5. Conclusions

Our study indicated that AMF could significantly alleviate Cd toxicity by enhancing the biomass, improving photosynthetic efficiency, and reducing the transfer of Cd to the shoot. However, when Cd concentration rises to 500 μM, the mycorrhizal efficiency is limited, the root colonization decreased, and the proportion of arbuscule decreased significantly, thus reducing the effect of alleviating Cd toxicity in *E*. *grandis*. From the cellular ultrastructure perspective, AMF retained Cd in the symbiotic cell so as to reduce the Cd toxicity to the root cell of *E*. *grandis*. This may be the mechanism by which AMF helps *E*. *grandis* resist heavy metal toxicity at the subcellular level. Our work helps to explain the mechanism of Cd toxicity alleviation by *R. irregularis* and provides a theoretical reference for using woody in heavy-metal-polluted soil remediation. Interestingly, hexagonal crystals with regular shapes were observed in our research. In the future, its composition and whether it is involved in AMF chelating heavy metals should be given more attention.

## Figures and Tables

**Figure 1 jof-09-00140-f001:**
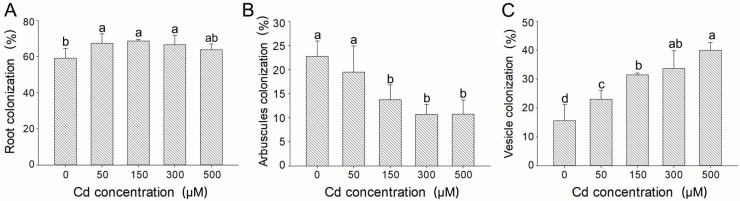
**AMF colonization status of *E*. *grandis* with AMF under different Cd concentrations.** Root colonization (**A**), arbuscules colonization (**B**), and vesicle colonization (**C**) of *E*. *grandis* inoculated with *R. irregularis* grown under different Cd concentrations (mean ± SE, *n* = 4). By ANOVA, both root colonization (**A**), arbuscules colonization (**B**), and vesicle colonization (**C**) were significantly (*p* < 0.05) affected by Cd concentration. The same lowercase letters above the columns indicate no significant (*p* < 0.05) differences among all treatments by Duncan’s multiple range test.

**Figure 2 jof-09-00140-f002:**
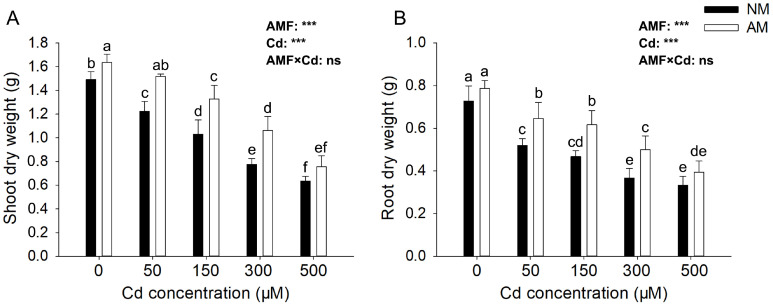
**Effects of AMF on the growth parameters under different Cd concentrations.** Shoot (**A**) and root (**B**) dry weight of E. grandis inoculated with/without *R. irregularis* grown under different Cd concentrations (mean ± SE, *n* = 4). AM and NM, respectively, represent inoculation with AMF and non-inoculation control. The same lowercase letters above the columns indicate no significant (*p* < 0.05) differences among all treatments by Duncan’s multiple range test. Significant effect of two-way ANOVA: *** *p* < 0.001; ns: no significant effect.

**Figure 3 jof-09-00140-f003:**
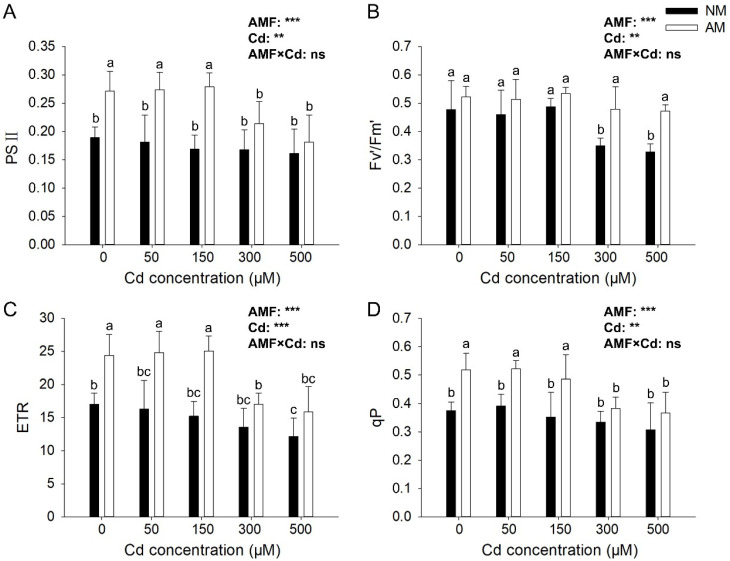
**Effects of AMF on photosynthesis under different Cd concentrations**. Actual photochemical quantum yield of ΦPSII (**A**), effective photochemical quantum yield of ΦPSII (**B**), electron transport rate (**C**), and photochemical quenching coefficient (**D**) of *E*. *grandis* inoculated with/without *R. irregularis* under different Cd concentrations (mean ± SE, *n* = 4). AM and NM, respectively, represent inoculation with AMF and non-inoculation control. The same lowercase letters above the columns indicate no significant (*p* < 0.05) differences among all treatments by Duncan’s multiple range test. Significant effect of two-way ANOVA: *** *p* < 0.001; ** *p* < 0.01; ns: no significant effect.

**Figure 4 jof-09-00140-f004:**
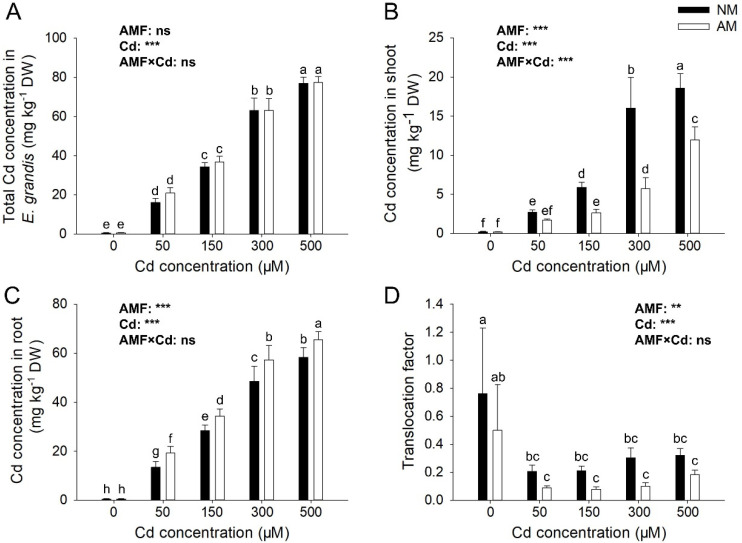
**Effects of AMF on the Cd absorption under different Cd concentrations.** Total Cd concentration in *E*. *grandis* (**A**), Cd concentration in shoot (**B**), Cd concentration in root (**C**), and translocation factor of Cd (**D**) of *E*. *grandis* inoculated with/without *R. irregularis* under different Cd concentration (mean ± SE, *n* = 4). AM and NM, respectively, represent inoculation with AMF and non-inoculation control. The same lowercase letters above the columns indicate no significant (*p* < 0.05) differences among all treatments by Duncan’s multiple range test. Significant effect of two-way ANOVA: *** *p* < 0.001; ** *p* < 0.01; ns: no significant effect.

**Figure 5 jof-09-00140-f005:**
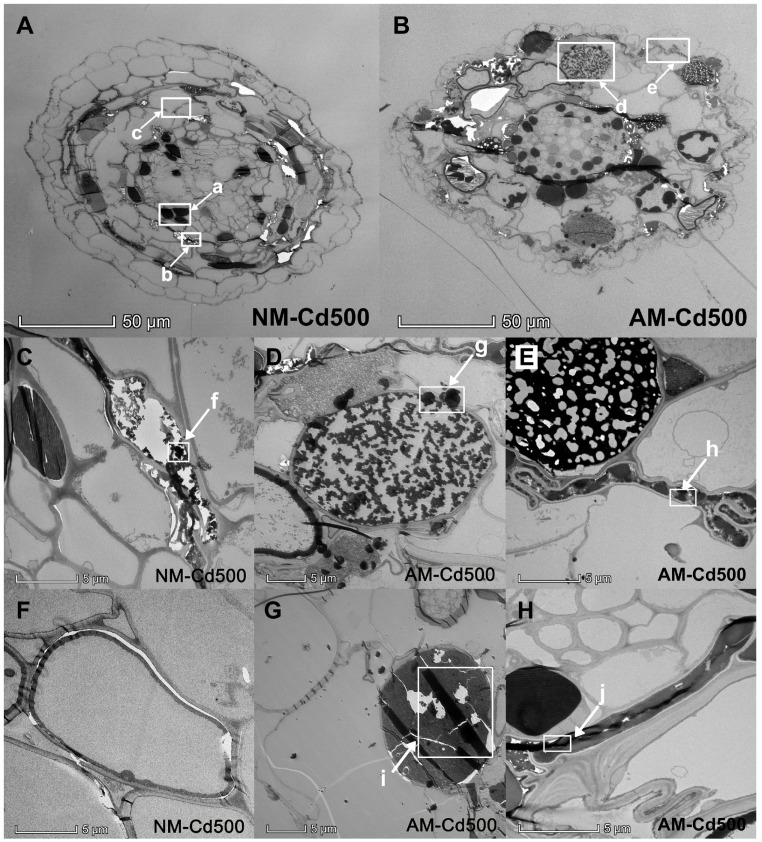
**Ultrastructure of non-mycorrhizal/mycorrhizal *E. grandis* root cells under TEM.** All sections were negatively stained with 2% uranyl acetate and 1% lead citrate. (**A**): the cross-section of non-inoculation *E. grandis* root cells; (**B**): the cross-section of inoculation *E. grandis* root cells; (**C**,**F**): the root cells in non-inoculation control; (**D**,**E**,**G**,**H**): the root cells inoculation with AMF. a, b: the Cd deposition site; c: the ruptured cell wall; d: the arbuscule of AMF; e: the hypha of AMF; f, i: the Cd deposition site; g: hexagonal crystals with regular shape; h, j: Cd deposition in hypha.

**Figure 6 jof-09-00140-f006:**
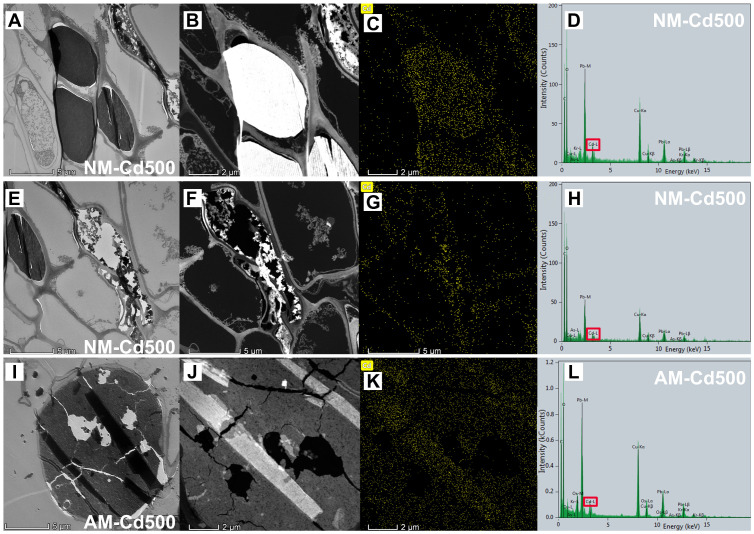
**Deposition of Cd in non-mycorrhizal/mycorrhizal *E. grandis* roots under TEM-EDS.** (**A**,**E**): Cd locations in NM *E. grandis* root by TEM; (**I**): Cd locations in AM *E. grandis* root by TEM; (**B**,**F**): Cd locations in NM *E. grandis* root by energy dispersive X-ray spectroscopy; (**J**): Cd locations in AM *E. grandis* root by energy dispersive X-ray spectroscopy; (**C**,**G**,**K**): the form of Cd in the energy spectrum; (**D**,**H**,**L**): the results of energy dispersive X-ray spectroscopy analysis; red box in (**D**,**H**,**L**): the peaks of Cd.

**Figure 7 jof-09-00140-f007:**
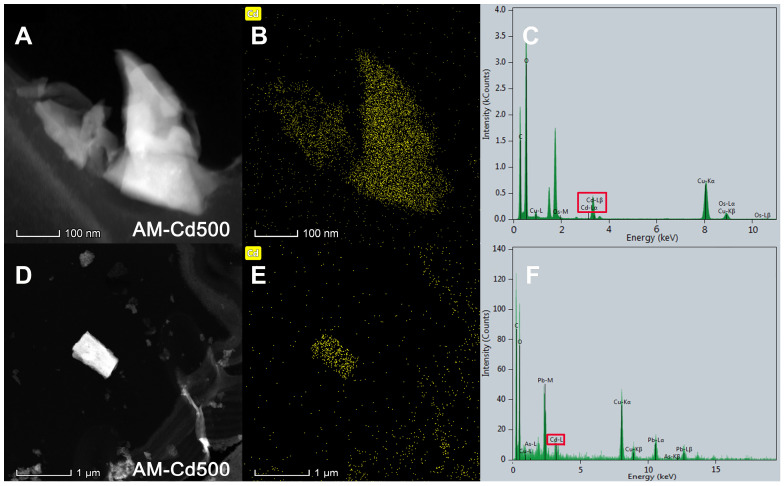
**Morphology of Cd in mycorrhizal *E*. *grandis* root cells under TEM-EDS.** (**A**,**D**): Cd locations in mycorrhizal *E. grandis* root by energy dispersive X-ray spectroscopy; (**B**,**E**): the form of Cd in the energy spectrum; (**C**,**F**): the results of energy dispersive X-ray spectroscopy analysis; red box in (**C**,**F**): the peaks of Cd.

## Data Availability

Not applicable.

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
