# Peer review of "Effects of Arbuscular Mycorrhizal Fungi on the Growth and Root Cell Ultrastructure of Eucalyptus grandis under Cadmium Stress"

_jof, 2023, doi:10.3390/jof9020140_

Round 1
Reviewer 1 Report
Introduction
There are scientific names that are misspelled, some have been modified so they must be updated.
The first time they are named, the full names must be written (as is the case with bacteria)
You have to improve your writing and English
Research objectives include the following: (1) to elucidate the mechanism by which AMF alleviates Cd toxicity in E. grandis from cellular ultrastructural and physiological perspectives; (2) to explore the distribution of Cd in control/mycorrhizal E. grandis roots, based on the results obtained the proposed objectives were not reached, the mechanisms involved in the responses cannot be explained. On the other hand, distribution and morphology of Cd in E. grandis root cells is shown in roots that are not known to be mycorrhized or not.
Materials and methods
The concentrations of Cd used are low and do not correspond to the rest of the literature. It is applied repeatedly but finally it is not known how much Cd is in the substrate because the concentration of Cd in the substrate is not determined. This information is important to add
It is not understood what is THE medium or THE nutritive solution, it must be explained and add quotes
Not all fluorescence parameters that appear later in the results are described, such as ETR, how are they obtained? Which indicates?
The same happens with the translocation factor, it must be explained in the methodology how it is obtained. If the Cd content in the substrate had been determined, other more interesting indices could be calculated to later decide if the plant can be used in a phytoremediation program.
Results
What it says in the text of figure 5 1 does not match with the heading of figure 5 1, in the text it says degraded arbusculos in 500 µM and in the figure it says clonization under non stress condition
In figure 1 C the letters of significance are wrong, b is missing
It would be convenient to put the scales the same to make it easier to compare the data.
The legend of figure 4 is confused, there are two a, two b, G, H, uppercase, lowercase, it is not understood
What is said about figure 4 C does not correspond
The legend of figure 5 is not understood
In figure 6 it is not clear if they are NM or AM roots, there could be both cases to be able to compare
Discussion
It is not appropriate to divide the section into subheadings
The discussion is very poor and the results found are not fully explained or justified.
Conclution
Conclusions are drawn regarding the efficiency of mycorrhization limited to 500 µM when in fact it is not known what concentration was reached in the substrate because it was never measured. This information is very important and cannot be missing.
Bibliography
There are very old quotes that should be updated
Reviewer 2 Report
This study focuses the mechanism by which AMF alleviates Cd toxicity in E. grandis and to explore the distribution of Cd in E. grandis roots. It is an interesting topic. However, minor revisions are still required before its final publication.
Specific comments
1. Line 171 - what about the case for 500 μM Cd?
2. Line 172 - what about minimum value?
3. The description for Figure 1 appears only on line 188. I suggest breaking Figure 1 into 2 figures: AMF colonization status of E. grandis under different Cd concentration / The growth parameters of E. grandis with AMF under different Cd concentration. Thus, the explanations to Figure 1 will not be so long, and thus not very legible.
4. Please format the references as required by the journal. https://www.mdpi.com/journal/jof/instructions
5. Please also add information about Supplementary Materials and Conflicts of Interest.
Round 2
Reviewer 1 Report
The corrections made to the manuscript are adequate. Regarding the concentration of Cd in the substrate, not everything that is added remains bioavailable or available to the plant. Vermiculite can adsorb Cd to its surface and decrease bioavailability. The pH also affects it and its value is not mentioned here. Not everything that is added can be absorbed by the plant. For this reason, the measurement of total and available Cd in the substrate is important.